# Soil Properties and Weed Dynamics in Wheat as Affected by Rice Residue Management in the Rice–Wheat Cropping System in South Asia: A Review

**DOI:** 10.3390/plants10050953

**Published:** 2021-05-10

**Authors:** Ramanpreet Kaur, Simerjeet Kaur, Jasdev Singh Deol, Rajni Sharma, Tarundeep Kaur, Ajmer Singh Brar, Om Parkash Choudhary

**Affiliations:** 1Department of Agronomy, Punjab Agricultural University, Ludhiana 141004, India; rbhullar905@gmail.com (R.K.); deoljs@pau.edu (J.S.D.); rajni-sharma@pau.edu (R.S.); tarundhaliwal@pau.edu (T.K.); braras@pau.edu (A.S.B.); 2Department of Soil Science, Punjab Agricultural University, Ludhiana 141004, India; opchoudhary@pau.edu

**Keywords:** burning, residue incorporation, mulching, organic carbon, insect pests, diseases, rodents

## Abstract

The rice–wheat cropping system (RWCS) has substantially contributed in making India self-sufficient in food grain production; however, rice residue management is of great concern, threatening the sustainability of this system. Rice residue is invariably disposed of by farmers through open burning. In addition to environmental pollution, residue burning of rice also leads to loss of soil nutrients. One of the alternatives to overcome these problems and sustain the RWCS is managing the rice residues in the field itself. Rice residue retention has variable effects on agricultural pests (namely, weeds, insect pests, diseases, and rodents) in the RWCS. High weed infestation in the RWCS results in high consumption of herbicides, which leads to several ecological problems and evolution of herbicide resistance. The shift from intensive tillage to conservation tillage causes major changes in weed dynamics and herbicide efficacy. Incorporation of rice residue reduces weed density and helps in improving soil physical, chemical, and biological properties. Rice residue retention on the surface or mulching reduces weed density and the biomass of both grass and broadleaf weeds in wheat crop as compared to its removal. Long-term field studies involving the use of rice residue as a component of integrated weed management strategies are needed to be done in the RWCS.

## 1. Introduction

A vast range of crops are cultivated in different agroecological regions of India, leading to production of huge amounts of crop residues. Crop residues are materials left on field after the crop has been harvested. There is wide variation in the generation of crop residue of different crops, e.g., rice (*Oryza sativa* L.), wheat (*Triticum aestivum* L. emend. Fiori et Paol.), corn (*Zea mays* L.), millets, sugarcane (*Saccharum officinarum* L.), cotton (*Gossypium arboreum* L.), jute (*Corchorus capsularis* L.), and pulses. This noneconomical part of the crop after harvesting the economical part is generated in huge volume, approximately 550 million tons (Mt) per annum [1]. The problem of crop residue burning is predominant in the rice crop, as it generates maximum on-farm residue as compared to other crops (Table 1 [2,3]). The maximum annual crop residue is generated in Uttar Pradesh followed by Punjab, Maharashtra, and West Bengal [2]. It is considered as waste, but it is a useful natural resource as it retains about 25%, 75%, and 50% of nitrogen and phosphorus, potassium, and sulfur, respectively [4]. There are several ways via which crop residue can be used such as cattle feed, composting, fuel, roof thatching, substrate for mushroom production, mulching, and biofuel [5,6,7,8,9]. However, a large portion of crop residue is burnt on farm to clear the field for sowing of succeeding crops. The problem of residue burning is mainly concentrated in the area under-irrigated agriculture, particularly in the rice–wheat cropping system (RWCS). Due to agriculture mechanization, along with labor shortage, the problem of residue burning has been increasing. An attempt is made to highlight different issues related to rice residue management in the intensive RWCS of northwestern India. These issues must be considered while formulating and implementing strategies for enhanced production and sustainability of the RWCS in south Asia.

### 1.1. Rice–Wheat Cropping System in Indo-Gangetic Plains of South Asia 

The RWCS is the dominant cropping system in the world, and it is grown on 24 million hectares (Mha), spread across China, Bangladesh, India, Nepal, and Pakistan [10]. The Indo-Gangetic plains of Bangladesh, India, Nepal, and Pakistan cover an area of 13.5 Mha, accounting for 33% of total rice area and 42% of total wheat area across these four countries [11]. Rice–wheat has been the dominant cropping system in Indo-Gangetic plains of India under assured irrigation for 55–60 years. In India, rice and wheat were grown on areas of 44.2 Mha and 29.3 Mha with a total production of 116.5 Mt and 103.6 Mt, respectively, during 2019–2020 [12]. The major RWCS-growing states in India are Uttar Pradesh, Punjab, and Haryana, which cover around 10 Mha of the total IGP regions [12]. The bumper production of rice and wheat crops during the green revolution led to self-sufficiency in food grains and improved the economic condition of the peasantry. There are many factors such as overexploitation of groundwater, crop residue management, formation of subsoil hard pan with a consequent increase in bulk density, multi-nutrient deficiencies, and weed infestation [13,14] that are threatening the sustainability of the RWCS.

Among all these sustainability issues of the RWCS, rice residue management is of great concern because disposal of rice residue has turned out to be a huge problem [7]. The harvesting of rice and wheat crops with a combine harvester leaves stubbles of about 30–40 cm in height. The loose straw is spread evenly in the field due to the attachment of a super straw management system to the combines [15,16]. Wheat straw is mainly salvaged by the farmers as cattle feed (Table 2 [17]); however, management of rice residue is a problem. Paddy straw contains 30–45% cellulose, 20–25% hemicellulose, and 15–20% lignin, thus restricting its use for fodder purpose [18]. Rice residue is invariably disposed of by farmers through open burning because of their anxiety to sow the following winter crops (wheat/field peas/potato) at the earliest convenience.

### 1.2. Ill Effects of Burning

Every year in October–November, a vast cloud of smoke engulfs the entire northwest India, as farmers burn the residue of freshly harvested rice crop in these states [1]. Rice residue burning results in both on-farm and off-farm extensive impacts such as air pollution, health problems, and losses of nutrients. The open burning results in emission of different ranges of harmful gases such as carbon monoxide/dioxide, nitrous/nitric oxide, sulfur dioxide, and methane (Table 3 [19,20,21,22]). These gases adversely affect the atmosphere and are hazardous to living organisms [23,24,25,26]. Residue burning causes various respiratory and cardiovascular diseases. Inhalation of fine particulate matter (PM) of less than 2.5 µg triggers asthma and can even aggravate symptoms of bronchial attack. A study conducted at Punjab reported that 60% of the population in Punjab living in rice-growing areas suffers from air pollution caused by residue burning [27].

Residue burning depletes the soil fertility, and approximately 400 kg of organic carbon, 5.5 kg of nitrogen, 2.3 kg of phosphorous, 25 kg of potassium, 1.2 kg of sulfur, and 50–70% of micro-nutrients are lost by burning one ton of paddy straw [16]. This loss of plant nutrients results in an extra expenditure of about 50.5 USD per hectare for replenishment of NPK alone [28]. Rice residue per se is not a problem; however, its burning is a real culprit, as it leads to collateral problems of environmental pollution, depletion of plant nutrient, and fodder scarcity. To overcome problems of burning, rice residue can be managed in field via various in situ and ex situ methods. In situ methods include straw incorporation in the field and its use as mulching material. In situ rice residue management methods help in increasing the nutrient value or soil fertility as compared to ex situ methods. The incorporation of paddy straw in the soil or mulching has favorable effects on soil physical, biological, and chemical properties [29].

## 2. Materials and Methods

Until now, isolated studies conducted on rice residue management at the field-scale level were restricted to problems in the fields of agronomy, soil science, agricultural engineering, crop protection, etc. We compiled and described relevant information on crop residue and its impact on soil health, environmental issues, pest dynamics, and weed management under the RWCS in this comprehensive review. The literature for this review was collected by searching various scientific electronic databases, including Google Scholar (https://scholar.google.com), Springer (https://link.spinger.com), Science Direct (https://sciencedirect.com), ResearchGate (https://www.researchgate.net/), Punjab Pollution Control Board (PPCB), Patiala, Punjab (http://www.ppcb.gov.in/index.aspx), Punjab Remote Sensing Centre (PRSC), Ludhiana, and Punjab (http://prsc.gov.in/?AspxAutoDetectCookieSupport=1) using the following keywords: “crop residues/rice residue and soil health/soil properties”, “happy seeder and wheat growth and productivity”, and “rice residue management and weed density”, considering studies written in English. Additional knowledge from other sources of literature was extracted (books and theses). Altogether, 115 references with respect to rice residue in the RWCS from 1999 to 2020 were chosen without bias.

A review on rice residue management under rice–rice cropping systems was published in 2020, which is a prominent cropping system in the subhumid and humid regions of southeast Asia [5]. We reviewed work related to soil properties and pest dynamics and management with emphasis on weeds in wheat crops as affected by rice residue management methods in the RWCS, which is dominant in subtropical and semiarid regions of south Asia. The review gives general information about the crop residue generation and ill effects of burning with respect to climate change and health hazards. The pest dynamics of wheat, especially weeds in different rice residue management scenarios, is studied. Lastly, different rice residue management approaches in fields for managing weeds are subcategorized.

## 3. Impact of Crop Residues on Soil Health

Maintenance of soil health is necessary for sustainable agriculture production. Residue retention on the soil surface or its incorporation into soil has several positive influences on the physical, biological, and chemical properties of soil. These practices modify soil structure and aggregate stability by increasing hydraulic conductivity and reducing bulk density, as well as building up soil organic matter and nitrogen reserves that enhance soil fertility. Residue also provides energy for growth and activities of soil microbes. 

### 3.1. Crop Residue vs. Soil Physical Properties

Soil has been degraded due to intensive crop cultivation and removal of crop residue over the years, resulting in lower carbon inputs into the soil [30], leading to an absence of mechanical protection that disperses the pressure caused by machine traffic [31] and additional wheel traffic over the field to collect and remove crop residue. To overcome these antagonistic effects of residue removal on soil health, managing crop residues in the field is a low-cost, simple, and sustainable approach. The retention and incorporation of residue has been reported to increase water-stable aggregates by 15.65% and 7.53% in the 0–15 and 15–30 cm soil layers, respectively [32]. Different soil hydro-physical properties such as infiltration, evaporation, bulk density, hydraulic conductivity, and porosity are modified by the presence of crop residue (Table 4 [33,34,35,36]). The application of crop residues was found to improve soil porosity by 30% [37] and aggregate stability by 15.2–21.0% [33,38], thereby facilitating better gas exchange, water transport, and soil organic carbon decomposition rates [39]. The in situ management of crop residues is a key factor for maintaining soil physical and hydraulic processes. Effective management of water, nutrient, and soil restoration due to crop residue retention will lead to sustainable production of the RWCS to meet the food requirements of an ever-expanding population.

### 3.2. Crop Residue vs. Soil Chemical Properties

Soil chemical properties, including adequacy of soil organic carbon and essential plant nutrients, are superior indicators of soil health and crop productivity. A very small profile (0–30 cm) of soil is actually used for production of the RWCS and, if it is not maintained and well nourished, it may undergo degradation or depletion [40]. Crop residues when retained in the field return nutrients to soils upon decomposition [41] and affect nutrient availability and its dynamics [20]. Returning residue to soil exerts favorable effects on several chemical properties of soil (Table 4).The incorporation of crop residue was found to increase the organic carbon by 33.3–40.9% and help in recycling of soil nutrients, thereby enhancing soil fertility and productivity [42,43,44]. The dynamics of organic carbon is essential for understanding its impacts on soil health [45]. The study of soil organic carbon status and other chemical properties is important for understanding the biological health of soil.

### 3.3. Crop Residue vs. Soil Biological Properties

Soil micro- and macro-fauna play a very important role in maintaining soil biological health. Biological attributes are more sensitive to soil management changes when compared to chemical and physical attributes [46]. Macro-fauna such as earthworms increase with increasing application of crop residues in soil. With addition of 5 t·ha^−1^ of residue, a 30% increase in earthworm population was reported [47]. The increased population of macro-fauna enhances the natural enemy activity and reduces the weed seed bank. Residue retention along with zero tillage facilitates the creation of permanent habitat of soil macro-fauna, including arthropods and rodents [48]. In contrast, the population of earthworms was reported to be affected by residue C:N ratio and the polyphenol concentration of crop residue. The C:N ratio and polyphenol concentration of crop residue are important factors in determining the palatability of residue, whereby a high C:N ratio and a high polyphenol concentration were found to be negatively correlated with microbial activity [49,50]. Soil micro-fauna depends upon soil organic carbon for metabolism [51]. The change in soil organic carbon content affects microbial population, composition, and function [52]. The microbial biomass plays an important role in nutrient cycling that helps in maintaining sustainability of the ecosystem [53]. There are several studies that quantified the effect of crop residue on soil biota, as listed in Table 4. It was seen that addition of crop residues could quickly improve the microbial activity, thereby improving the ecological environment of the soil itself.

## 4. Impact of Crop Residues on Agricultural Pests 

Agricultural pests are the major determinant for adoption of any new technology. Rice residue in the field harbors many pests, and crop residue retention has variable effects on different agricultural pests such as weeds, insects, diseases, and rodents.

### 4.1. Weeds

Weeds are unwanted and undesirable plants that interfere with the utilization of the land and water resources, thus adversely affecting crop production and human welfare (Buchholtz 1967). Twelve weeds, namely, littleseed canarygrass (*Phalaris minor* Retz.), burclover (*Medicago denticulata* L.), oat (*Avena ludoviciana* Durieu), lambsquarters *(Chenopodium album* L.), chicory (*Cichorium intybus* L.), dock toothed (*Rumex dentatus* L.), sweetclover (*Melilotus indicus* (L.) All), *Melilotus albus* Medik, swinecress (*Coronopus didymus* (L.) Sm.), field bindweed (*Convolvulus arvensis* L.), common vetch (*Vicia sativa* L.), and ragweed (*Parthenium hysterophorous* L.), were associated with wheat crop in India [54]. Weed monitoring and surveys were conducted to detect the appearance of new weed species from 2008–2017, leading to the novel identification of false cleavers (*Galium spurium* L.), henbit (*Lamium amplexicaule* L.), burweed (*Soliva sessilis* Ruiz & Pav.), evening primose *(Oenothera laciniata* Hill), and desert tobacco (*Nicotiana plumbaginifolia*) infesting the wheat fields in Punjab, India [55]. The infestations of little mallow (*Malva parviflora* L.), pigweed *(Amaranthus retroflexus* L.), bluegrass (*Poa annua* L.), rabbitfoot (*Polypogon monspliensis* (L.) Desf.), and *C. didymus* are increasing in the RWCS of the region [56]. 

Depending on the type of weed, its intensity, period of infestation, crop competition, and climatic conditions, the losses caused may vary. Weeds pose a serious threat to wheat production and account for 20–40% of yield losses [57,58]. Yield loss up to 35% was reported in wheat crop due to a mixed infestation of *P. minor, Avena ludoviciana,* pimpernel scarlet *(Anagallis arvensis* L.), and *C. album* [59]. There is regional variation in yield loss due to weeds in the RWCS (Table 5 [60,61,62,63,64]). *Phalaris minor* is a highly competitive weed of wheat and can cause yield reductions of up to 95% [64,65]. *Phalaris minor* density of 25, 50, 100, 200, and 400 plants·m^−2^ resulted in yield reductions to the tune of 12%, 25%, 41%, 59%, and 71%, respectively [66], and 250 plants·m^−2^ of wild oat (*Avena fatua* L.) caused yield loss of up to 40% [67]. Weeds in wheat may result in complete crop failure, and weed management through herbicide use is one of the most important components of crop production [68].

The higher cost of labor for manual weeding has restricted its use, and herbicides are widely used by farmers [69]. The continuous use of the same herbicides over the years has led to the evolution of a resistant biotype in some weeds in the RWCS. For example, isoproturon was recommended in 1977 for the control of *P. minor* in wheat [70]. However, extensive use of isoproturon led to its resistance in *P. minor* and, consequently, there was decline in wheat production in 1995 [65,68]. Thereafter, clodinafop and sulfosulfuron effectively controlled the isoproturon-resistant *P. minor* and improved wheat productivity [68]. The problem of *P. minor* worsened after the evolution of multiple resistance against clodinafop, fenoxaprop, and sulfosulfuron. It is clear from this scenario that there is a need to change weed management tactics and strategies for sustainable weed management. 

Use of paddy straw has the potential to sustainably manage the weeds at low cost. Combined adoption of multiple weed control options, both chemical and nonchemical practices such as residue management (retention or incorporation), can help in the effective management of weeds in wheat. Various rice residue management strategies have a variable effect on weed dynamics in wheat due to either physical hindrance or alleopathic interactions.

### 4.2. Other Agricultural Pests

#### 4.2.1. Rodents

Rodents are the major pests in our agro-ecosystem, causing direct or indirect losses to crops in our fields [71]. The attack of rodents varies with season, crop, burrow density, etc. The presence of rats was examined on the basis of the presence of burrows they made in fields. Residue retention in wheat crop had the maximum burrow density (60 burrows per hectare) causing 28% damage [72]. In a survey conducted with farmers in northwestern India, 28.6% of respondents recognized rodent damage as a major problem with high intensity, while 57% of respondents considered this problem of moderate severity in residue-retained wheat crop fields [73]. Moreover, an additional expenditure of 5 USD per hectare for control of rodents has been reported in residue-retained wheat crop as compared to wheat crop sown without residue [74]. To control damage by rodents, proper management measures including sanitary and chemical approaches need to be adopted during spring and winter. Systematic research efforts need to be carried out to quantify the impact of residue management on different soil flora and fauna in the RWCS.

#### 4.2.2. Insect Pests

The pest scenario of wheat has changed with the presence of residue in field. With the adoption of residue management strategies, new insect pests such as pink stem borer (*Sesamia inferens* Walker) and army worm (*Mythimna separata* Walker) have emerged as major pests of wheat crop in northern India. Sporadic pests have become the major pests because of a shift from conventional to conservational agriculture. The life cycle of pests is disturbed under conventional agriculture, whereas pests remain in straw in residue-retained fields, allowing the population to build quickly under favorable conditions. The information gathered from farmers indicated that the pest problem in wheat crop sown with residue retention is moderate to severe in 43% of cases [73]. The higher pest problem was ascribed to carry-over of insects in leftover residue in the field. The incidence of pink stem borer was the lowest in conventional sowing without residue (0.05%) as compared to residue-retained wheat crop (0.25%) [75,76].The problem of army worm in wheat was observed earlier in March–April during the heading stage. However, in recent years, its damage has been observed in December. This might be due to the presence of loose straw in the field, which acts as a shelter for infesting larvae [77]. Farmers are facing difficulty in controlling insect pests in residue-retained fields, and the change in timing of pest occurrence or maximum pest damage has further escalated this problem. Moreover, an additional expenditure of 8.6 USD per hectare as a cost of insecticide is involved in residue-retained wheat crop as compared to wheat sown without residue [74]. There is a need to explore the problem of these pests under changed scenarios, and a systematic study should be conducted to reinvestigate the life cycle of these insect pests under different residue management systems.

#### 4.2.3. Diseases

Crop residues play a prominent role as the primary infection source of many soil-borne diseases in the succeeding crops [78]. Cultural control of soil-borne diseases involves the removal of crop residue from the field. The incidence and the severity of Fusarium head blight of wheat were higher than 12% and 30%, respectively, under residue-retained fields as compared to its incorporation with moldboard plough [79]. The infection potential of crop residue depends on the disease severity and control measures taken in the previous crop. In the RWCS, inoculum of several soil-borne diseases (sheath blight, loose smut, etc.) may survive in the soil. Therefore, extensive field trials need to be conducted to study the implications of residue management methods on carry-over of disease to the succeeding crops. 

## 5. Use of Rice Residue for Weed Management

Weed dynamics in wheat is highly influenced by rice residue management strategies. The weed dynamics is affected by different residue management methods such as burning, incorporation, and mulching (Figure 1).

### 5.1. Burning

Burning of crop residues has many disadvantages, such as loss of many essential nutrients, release of harmful gasses, and high soil temperature, but it has been found that burning of crop residue can be used as one of the methods for controlling weeds by reducing the surface seed banks of many weeds. Crop residues upon burning can produce sufficient heat to kill weed seeds in the upper soil layer of 0–1 cm. Weed seeds and density were found to be lower in residue burned plots due to loss of seed viability [80]. 

Various methods used for managing weed population are complete burning, partial burning, narrow windrow burning, etc. Different methods of burning act differently for controlling weeds. Seeds close to the soil surface are more likely to be killed than weed seeds that are buried deep in the soil profile. A study conducted in Madhya Pradesh, India revealed that burning or incorporation of rice crop residue significantly reduced *P. minor* in wheat crop as compared to its removal, but density of *M. denticulata* was not affected by different rice residue methods [81]. However, in another study conducted in Haryana, India, the plant population of *P. minor* was higher to the tune of 31.2–39.2% in plots where burning of residue load of 6–12 t·ha^−1^ was done as compared to its removal. This increase in *P. minor* population with residue burning was due to the fact that seed germination was stimulated by higher temperature or smoke during burning [82]. Partial burning of crop residue resulted in a reduction in weed density and biomass by 21.5% and 18.7%, respectively, as compared to the conventional sowing method [83].

Although open burning of crop residue helps reducing weed density, it also results in a huge loss of carbon and other nutrients that are concentrated in these crop residues. Field studies conducted in Australia reported that windrow burning is more effective in killing weed seeds present on the soil surface than burning standing stubbles in the open. Windrow burning acts as a weed seed destruction method, whereby the temperature at the soil surface exceeds 400–500 °C, which was found to return back to ambient levels in just 1–3 min [84]. This high temperature in a narrow strip was sufficient to kill weed seeds without harming soil physical, biological, or chemical properties. In a similar study conducted in the US, it was observed that narrow row burning of soybean residues resulted in mortality or loss of viability of seeds of evaluated weed species [85]. Narrow windrow burning may be recommended as a strategy for weed management, provided it is legal within the permissible limits of the farmers’ fields. The positive or negative influences of burning on soil microbial activity and fertility status need to be systematically studied. Again, the implications of burning for the management of the weed seed bank are well known, but additional research is needed to fully understand how best to integrate such a strategy into the RWCS and its sustainability.

### 5.2. Incorporation

Farmers have two alternatives in combine-harvested fields: in situ use or burning. Use of rice residue as a weed management method is a sustainable way to manage residue, as well as weeds, in wheat. However, this energy-intensive operation of crop residue incorporation is not preferred by most farmers, and it was reported to have variable effects on crop yield. Wheat grain yield with residue incorporation was lowered by 9.7% due to nitrogen immobilization as compared to residue burning [86]. In contrast, it was reported that incorporation of residue did not adversely affect wheat crop yield; rather, it helped in increasing the soil organic carbon and total N by 23.7% and 9.8%, respectively, as compared to residue removal [87].

Incorporation of crop residues is mostly done by tillage operations with different implements such as a cultivator followed by rotavator, moldboard plough followed by a rotavator, and a rotavator alone. All these methods have different effects on weed flora and its distribution in the soil profile. Reductions of 32.6% in weed density and 31.3% in weed biomass were recorded due to rice residue incorporation as compared to its removal [81]. Weed density was lowered by 18.9% with incorporation of residue compared to when residue was removed [88]. Incorporation of residue (8–10 t·ha^−1^) resulted in a reduction in weed population of 30.4–37.7% as compared to residue burning [89]. In contrast, weed biomass in residue-incorporated (5 t·ha^−1^) fields was increased by 6–12% as compared to straw removal or its burning [63]. Rice straw incorporation helps with a reduction in weed density or biomass. In contrast, some studies observed that incorporation has the lowest effect on weed suppression. The effect of incorporation of crop residue on weed dynamics needs to be studied further under long-term field research trials.

### 5.3. Surface Mulching

Residue retention on the surface as mulch is more beneficial than its incorporation, and it can lead to low weed seed germination and weed smothering. Weed dynamics is significantly affected by residue retention on the surface. Residue helps to reduce weed seed emergence by avoiding exposure to light and through mechanical impedance of the weed seedlings [90]. Residue may be retained in the field either by spreading residue on soil surface or by chopping and spreading standing stubble along with loose straw of previously combine-harvested crop.

The shift from intensive tillage to residue retention with zero tillage in wheat resulted in a shift in weed flora The emergence of *P. minor* was lowered by 25–50% in residue-retained wheat crop fields than residue-removed fields under RWCS [91,92,93,94]. However, germination of some broadleaved weeds, such as *R. dentatus*, was almost doubled in residue-retained fields [92]. In a pot study, prosulfocarb plus metolachlor (3.75 kg·ha^−1^) resulted in complete mortality of junglerice (*Echinochloa colona* (L.) Link)*,* but seedling emergence was 33% and 41.3% under sorghum residue loads of 3 t·ha^−1^ and 6 t·ha^−1^ (field equivalent rate), respectively [95]. Rice straw mulching (7.5 t·ha^−1^) resulted in 34.4% and 7.1% reductions in the density of broadleaved and grass weeds, respectively, which were further reduced to 80.6% and 67.1% with a straw load of 20 t·ha^−1^ [96]. This reduction in weed population due to rice straw mulch was partly due to mechanical impedance and the presence of certain phytotoxic compounds in rice straw.

The load of retained residue is more important for considering it as a weed management approach. Weed seed germination was suppressed by 30.5% under 7 t·ha^−1^ residue on the soil surface [97]. A rice residue load of 2 t·ha^−1^ did not result in adequate weed control in wheat crop [98]; however, residue retention of 5–7.5 t·ha^−1^ resulted in a reduction in weed biomass of 23.4–44.1% as compared to complete removal of residue [82].

Therefore, the effect of residue as mulch is variable for different weed species and depends on the residue load. Retention of crop residue on the surface may protect weed seeds from predation and physical decomposition [99], while microbial decomposition may be increased in the presence of crop residues [100]. Furthermore, uniform coverage of mulch material (crop residue/straw) on the soil surface is a must for impeding light penetration and reducing weed seed germination. Long-term intensive research trials need to be conducted to ascertain the effect of different levels of residue load on different weed species in RWCS.

### 5.4. Allelopathic Potential of Rice Residue

Residue incorporation helps in improving soil health and suppressing the weed population because of the toxic effect of rice residue on weeds. Allelochemical studies performed on paddy straw demonstrated the occurrence of several metabolites that lead to an allelopathic effect on weeds. Paddy straw is a rich source of phenols such as *p*- salicyclic acid, *p*-coumaric acid, vanillic acid, syringic acid, ferulic acid, and mandelic acid [101].

Laboratory studies were conducted to see the allelopathic effect of rice straw for weed management in wheat. In a laboratory study, germination of *P. minor* seeds was reduced by 60.0–73.3% in the presence of 2 g of rice straw per 100 mm petri plate as compared to control (no straw addition). Furthermore, germination of seeds of *M. denticulata* was reduced by 60–80% in the presence of 2 g of rice straw as compared to control [102]. The presence of phenol content in rice residue leads to an inhibitory effect on weed seed germination and seedling growth. Similarly, in another laboratory study, the germination of *P. minor* was suppressed by 32.5–68.3% with 5% aqueous extract of paddy straw of different varieties [11]. Similarly, allelochemicals namely momilactones are released from the hulls of rice which suppressed the germination of Chinese sprangletop (*Leptochola chinenesis* (L.) Nees), *A. retroflexus* and smallflower umbrella sedge (*Cyperus difformis* L.) [103]. These laboratory studies show the potential of rice residue to control the weeds via allelopathic interactions under controlled conditions. More extensive field studies are needed to ascertain the content of allelochemicals in rice straw of different varieties and their role in weed management in situ and as bioherbicides at a field scale.

### 5.5. Effect of Crop Residues on Herbicide Efficacy

Retaining crop residue on the soil surface results in many advantages such as enhancing soil health, improving crop production, and managing weeds. Use of rice residue is an alternative sustainable approach for weed control in wheat under the RWCS. Various pre- and post-emergence herbicides are used for weed management in the RWCS. However, the efficacy of pre-emergence herbicides is reduced by retaining crop residue on the soil surface [104]. The presence of crop residue on the surface results in more interception of herbicides on the surface, thus inhibiting it from reaching the soil surface. Crop residue can intercept up to 80% of the applied pre-emergence herbicide, and this results in the reduced efficacy of the herbicide. This adsorbed herbicide on the residue surface is subjected to various losses such as leaching, photodegradation, and microbial degradation [105]. The efficacy of prosulfocarb plus metolachlor (3.75 kg·ha^−1^) for the control of *E. colona* was reduced as biomass increased by 46–70% in the presence of a field equivalent rate of sorghum residue load of 3–6 t·ha^−1^ [96]. Various field studies involving residue management methods and herbicides concluded that rice residue along with post-emergence herbicide helps in lowering the weed population [106]. A rainfall event is required within 7 days for leaching of pre-emergent herbicide from the residue, and herbicides that leach easily from the residue should be preferred for weed control in residue-retained fields [107]. Extensive efforts are required to study the effect of the amount of crop residue on the soil surface on the performance of pre-emergence herbicides. To achieve adequate weed control in the presence of residues, spray technology for the application of pre-emergence herbicides may be revalidated. In future studies, the feasibility of more efficient herbicides with new formulations and a lower interception rate in the presence of crop residue should be evaluated. 

## 6. Impact of Crop Residue Retention on Crop Growth and Yield

Soil health is improved with residue retention, which results in an improvement of crop growth and yield. However, a temporary delay in wheat emergence can be seen due to the presence of residue above the seed, which can be explained by the physical impedance of coleoptile growth [108]. In laboratory studies, the effect of different residue (corn, sorghum, and alfalfa) on the growth of wheat seedling was studied, and it was observed that wheat seed germination was reduced by 1–5% and shoot length was reduced by 15.9–41.3% due to the presence of 50 g of residue per 100 mm petri dish [109]. The effect of residue on plant height was variable, and substances (such as phenols, vanillic acid, syringic acid, ferulic acid, and mandelic acid) released from the residue inhibited seedling growth [110]. However, there was an increase in plant height by 8–11.8% with residue retention as compared to residue removal under field studies [45]. Non-decomposed residue with a high C:N ratio may adversely affect seed germination and plant growth [34]. However, residues have shown a yield-enhancing effect after decomposition, and an increase in wheat productivity to the tune of 14.8–18.6% was witnessed with crop residue retention as compared to residue burned or removed [111]. Wheat yield under zero tillage with residue retention was higher by 26.7% and 12.8% than bed-planted wheat and conventionally tilled wheat crop, respectively [112]. Residue retention is not widely adopted by farmers due to a number of associated problems such as pest problems, as well as the nonavailability of an expensive seeder and tractor with high-power traction for small and marginal farmers [73]. There is a need for field studies to identify sustainable farm mechanization solutions for direct seeding of wheat in residue-retained fields [113] and to tackle the pest problem. Rice residue management has multiple benefits such as reduced air pollution, water saving, weed suppression, and higher yield, along with buffering of temperature in the subsequent crop [114]. There is a need to change the perception of farmers for conservation tillage practices in wheat crop by increasing their awareness.

## 7. Conclusions

The labor-, water-, capital-, and energy-intensive RWCS of the south Asia has become less profitable. A huge amount of rice residue is generated every year in the RWCS, and this is a rich source of nutrients which helps in maintaining soil health, thus ultimately leading to better crop growth. Economically, it is easier to burn the residue in the field than using costly and energy-intensive machinery for its management. Farm machinery solutions for in situ crop residue management, raising awareness about the ill effects of crop residue burning, and empowering farming stakeholders are crucial steps for rice residue management. The sectorial thinking of the government should be avoided. Rather, a higher-level integrated approach is needed for in situ crop residue management. A paradigm shift in crop residue management is required for sustainability of the intensively cultivated RWCS. The in situ management of crop residues is the only sustainable approach to crop production and weed management. The shift from a conventional to conservational tillage system may manifest pest problems and cause a shift in incidence of pests such as weeds, insect pests, diseases, and rodents. 

Weeds pose the maximum threat to crop productivity and increase the cost of cultivation compared with other pests. For realizing the full genetic yield potential of the crop and sustaining food grain production to feed an ever-increasing population, weed management is essential. The continuous use of only one weed management strategy (herbicides) leads to a shift in weed flora, an increase in residual toxicity, and the development of herbicide resistance in weeds [115]. Therefore, these situations have forced researchers to investigate economically viable, ecologically sustainable, and technologically feasible options for weed management in wheat crop. Residue retention on the soil surface suppresses weed flora in the RWCS via mulching effects through mechanical impedance of the weed seedlings and by avoiding exposure to light. The allelopathic effects of rice residue may help in controlling weeds in wheat crop. Crop residue retention on the soil surface leads to adverse effects on the efficacy of pre-emergence herbicides. The role of crop residue management in achieving sustainable food production from the RWCS has become more important than ever. This article reviewed the effect of rice residue management on the soil properties (physicochemical and biological), pest dynamics, and grain yield of wheat crop in the RWCS. Crop residue management techniques are highly site-specific. Therefore, an integrated approach is required to formulate different solutions for the management of rice residue and for improving the profitability and sustainability of the intensively cultivated RWCS system.

## Figures and Tables

**Figure 1 plants-10-00953-f001:**
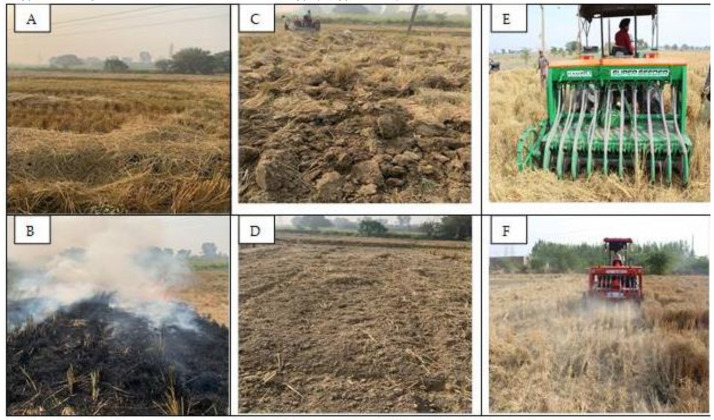
Photograph from fields showing (**A**) residue (loose and standing) after combine harvesting of rice crop, (**B**) residue burning, (**C**) residue incorporation with moldboard plough, (**D**) residue incorporation with moldboard plough and rotavator, (**E**) residue incorporation with superseeder, and (**F**) residue retained on soil surface as mulch.

**Table 1 plants-10-00953-t001:** Residue generation by different crops in India.

Crop	Residue Generated (Mt·Year^−1^)	Composition
Rice	122–231.9	Straw, husk
Wheat	110–130	Straw
Maize	71	Stover, husk
Millets	26	Straw
Sugarcane	107.5–141.0	Trash, bagasse, pressmud
Fibre (jute, mesta (*Hibiscus spp*), cotton)	80.0–122.4	Trash, sticks
Pulses	28	Stover

Source: [2,3].

**Table 2 plants-10-00953-t002:** End use of stubble by the farmers of Punjab, India.

End Use	Rice (% of Total Stubble Production)	Wheat (% of Total Stubble Production)
Fodder	7	45
Soil incorporation	1	<1
Burnt	81	48
Rope making	4	0
Miscellaneous	7	7

Source: [17].

**Table 3 plants-10-00953-t003:** Pollutants released by open burning of 1 ton of rice residue.

Pollutants (Gg)				References
Methane	Carbon Monoxide	Nitrous Oxide	Nitrogen Dioxide	
110	2306	2	84	[19]
1.33	113	-	8.6	[20]
680	2300	-	960	[21]
102	2138	2.2	78	[22]

**Table 4 plants-10-00953-t004:** Effect of crop residue retention on physical, chemical, and biological properties of soil.

Soil Properties	Impact of Crop Residue Retention	Reference
**Physical properties**		
Infiltration	(+) 20.6%	[33]
Bulk density	(−) 6.0%	[34]
Porosity	(+) 18.7%	[33]
**Chemical properties**		
Organic matter	(+) 18.0%	[35]
Organic carbon	(+) 43.9–66.7%	[32,34]
N, P, K	Increases	[34]
EC	Increases	[34]
**Biological properties**		
Microbial biomass	(+) 90–95%	[34]
Microbial activity bacteria Fungi	(+) 5–10 times (+) 1.5 to 11 times	[36] [36]

(+): increase; (−): decrease.

**Table 5 plants-10-00953-t005:** Yield loss due to different weed species infesting wheat in different regions.

Region	Yield Loss (%)	Species	Reference
**Punjab**	18–34	*Phalaris minor, Polygonum monospeliensis, Poa annua, Medicago denticulata, Anagallis arvensis,* cress garden *(Lepidium sativum* L.)*, Malva neglecta*	[60]
	60–70	*Phalaris minor, Polygonum monospeliensis, Poa annua, Medicago denticulata, Anagallis arvensis, Malva neglecta*	[61]
**West Bengal**	24–32	*Polygonum orientale, P. pensylvanicum, P. persicaria,* chickweed (*Stellaria media* (L.)) Vill.*,* diamond flower *(Oldenlandia diffusa* L.)*,* pennywort *(Hydrocotyl ranunculoides* L.f.)*,* groundcherry *(Physalis minima)*	[62]
**Karnal**	25–60	*Coronopus didymus, A. arvensis, Melilotus indicus* (L.) All, *Medicago denticulata, Rumex dentatus,* peavine *(Lathyrus aphaca* L.)	[64]
**Haryana**	15–50	*Phalaris minor, Polygonum monospeliensis, Coronopus didymus, A. arvensis, Melilotus indica, Medicago denticulata, Rumex dentatus, Lathyrus aphaca*	[63]

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
