# Peer review of "Soil Properties and Weed Dynamics in Wheat as Affected by Rice Residue Management in the Rice–Wheat Cropping System in South Asia: A Review"

_plants, 2021, doi:10.3390/plants10050953_

Round 1

Reviewer 1 Report

The present Review deals with the effects of different management of residues of rice crops on soil, and weeds. Although the argument is very interesting, a very low contribution to the current literate is provided since plenty of studies deal with the contribution of residues management worldwide: soil quality improvement, lower presence of weeds, GHG emissions due to in-field burning, etc... Furthermore, similar review studies have been published recently

https://link.springer.com/article/10.1007%2Fs10668-019-00370-z

https://www.tandfonline.com/doi/full/10.1080/03650340.2019.1661994

https://www.sciencedirect.com/science/article/pii/S2095311916613370?via%3Dihub

Besides, the “Methodology of the search” is missing within the manuscript which is very important. For example “We performed a literature search in, Scopus, and Google Scholar (up to April 30, 2020) using the following search terms in the title and abstract…” highlighting the importance of the Review and the lack in the literature that the authors aim to fill.  

Detailed comments:

  • Table 1: please report the unit of time. I guess it is Mt y-1. Being a review article, Authors should include references (possibly more than one) per crop to prove that numbers are correct
  • Line 96-100: “per se” italicus
  • Line 97: “As,” change into Since

 ..

Author Response

Word document attached

Reviewer 2 Report

Dear Authors, you should address my comments highlighted across the manuscript.

Author Response

Response attached in word document

Round 2

Reviewer 1 Report

The Authors improved the paper sufficiently

Author Response

I want to take this opportunity to thank you for reviewing and helping in improving the manuscript.

Reviewer 2 Report

Dear Authors, you have appropriately addressed my comments and, therefore, the manuscript can be accepted for publication, in my opinion.

Author Response

(The authors gave the same response as above.)
